# Watching the World Go By: Representation Learning from Unlabeled Videos

## Abstract

Recent unsupervised representation learning techniques show remarkable success on many single image tasks by using instance discrimination: learning to differentiate between two augmented versions of the same image and a large batch of unrelated images. Prior work uses artificial data augmentation techniques such as cropping, and color jitter which can only affect the image in superficial ways and are not aligned with how objects actually change e.g. occlusion, deformation, viewpoint change. We argue that videos offer this natural augmentation for free. Videos can provide entirely new views of objects, show deformation, and even connect semantically similar but visually distinct concepts. **We propose Video Noise Contrastive Estimation, a method for using unlabeled video to learn strong, transferable, single image representations.** We demonstrate improvements over recent unsupervised single image techniques, **as well as over fully supervised ImageNet pretraining**, across temporal and non-temporal tasks.[1]

## 1 Introduction

The world seen through our eyes is constantly changing. As we move through the world, we see much more than a single static image: objects rotate revealing occluded regions, deform, the surroundings change, and we ourselves move. Our internal visual systems are constantly seeing temporally coherent images. Yet many popular computer vision models learn representations which are limited to inference on single images, lacking temporal context. Representations learned from static images are inherently limited to an understanding of the world as many unrelated static snapshots.

This is especially true of recent unsupervised learning techniques (Bachman et al., 2019; Chen et al., 2020a; He et al., 2020; Hénaff et al., 2019; Hjelm et al., 2019; Misra & Maaten, 2020; Tian et al., 2019; Wu et al., 2018), all of which train on a set of highly-curated, well-balanced data: ImageNet (Deng et al., 2009). Scaling up these techniques to larger, less-curated datasets like Instagram-1B (Mahajan et al., 2018) has not provided large improvements in performance (He et al., 2020). Only so much can be learned from a single image: no amount of artificial augmentation can show a new view of an object or what might happen next in a scene. This dichotomy can be seen in Figure 1.

In order to move beyond this limitation, we argue that video supplies significantly more semantically meaningful content than a single image. With video, we can see how the world changes, find connections between images, and more directly observe the underlying scene. Prior work using temporal cues has shown success in learning from unlabeled videos (Misra et al., 2016; Wang et al., 2019; Srivastava et al., 2015), but has not been able to surpass supervised pretraining. On the other hand, single image techniques (Gutmann & Hyvärinen, 2010) have shown improvements over state-of-the-art by using Noise Contrastive Estimation (NCE). In this work, we merge the two concepts with Video Noise Contrastive Estimation (VINCE), a method for using unlabeled videos as a basis for learning visual representations. Instead of predicting whether two feature vectors come from the same underlying image, we task our network with predicting whether two images originate from the same video. Not only does this allow our method to learn how a single object might change, it also enables learning which things might be in a scene together, e.g. cats are more likely to be in videos with dogs than with sharks. Additionally, we generalize the NCE technique to operate on multiple positive pairs from a single source. To facilitate this learning, we construct Random Related Video

---

[1]Code and the Random Related Video Views dataset will be made available.

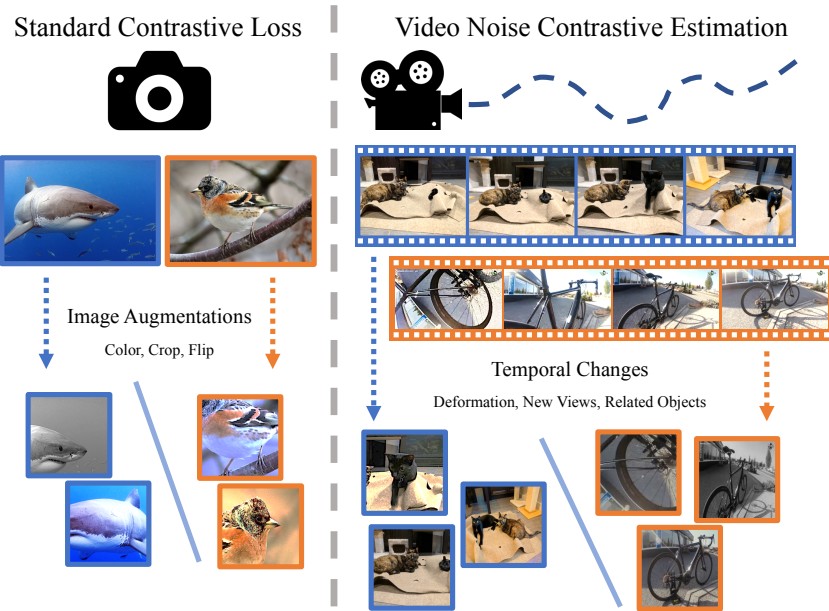

Figure 1: **NCE vs VINCE.** The standard contrastive setup learns to separate artificial augmentations of the same image. Our method uses novel views and temporal consistency which single images cannot provide.

Views (R2V2), a set 960,000 frames from 240,000 uncurated videos. Using our learning technique, we achieve across-the-board improvements over the recent Momentum Contrast method (He et al., 2020) as well as over a network pretrained on supervised ImageNet on diverse tasks such as scene classification, activity recognition, and object tracking.

## 2 RELATED WORK

### 2.1 NOISE CONTRASTIVE ESTIMATION (NCE)

The NCE loss (Gutmann & Hyvärinen, 2010) is at the center of many recent representation learning methods (Bachman et al., 2019; Chen et al., 2020a; He et al., 2020; Hénaff et al., 2019; Hjelm et al., 2019; Misra & Maaten, 2020; Tian et al., 2019; Wu et al., 2018). Similar to the triplet loss (Chechik et al., 2010), the basic principle behind NCE is to maximize the similarity between an anchor data point and a positive data point while minimizing similarity to all other (negative) points.

A challenge for using NCE in an unsupervised fashion is devising a way to construct positive pairs. Pairs should be different enough that a network learns a non-trivial representation, but structured enough that the learned representation is useful for downstream tasks. A standard approach used by (Bachman et al., 2019; Chen et al., 2020a; He et al., 2020) is to generate the pairs via artificial data augmentation techniques such as color jitter, cropping, and flipping. Contrastive Multiview Coding (Tian et al., 2019) uses multiple "views" of a single source image such as intensity (L), color (ab), depth, or segmentation, training separate encoders for each view. PIRL (Misra & Maaten, 2020) uses the jigsaw technique (Noroozi & Favaro, 2016) to break the image into non-overlapping regions and learns a shared representation for the full image and the shuffled image patches. Similarly, Contrastive Predictive Coding (CPC) (Hénaff et al., 2019) uses crops of an image as "context" and predicts features for the unseen portions of the image. We provide a more natural data augmentation by using multiple frames from a single video. As a video progresses, the objects in the scene, the background, and the camera itself may move, providing new views. Whereas augmentations on an image are constrained by a single snapshot in time, using different frames from a single video gives entirely new information about the scene. Additionally, rather than restricting our method to only use two frames from a video, we generalize the NCE technique to use many images from a single video, resulting in more computational reuse and a better final representation (Bachman et al. (2019) similarly makes multiple comparisons per pair, but each anchor has only one positive).

## 2.2 Unsupervised Learning Using Video Cues

In contrast with supervised learning which requires hand-labeling, self-supervised and unsupervised learning acquire their labels for free. These techniques create datasets which are orders of magnitude larger than comparable fully-supervised datasets. Whereas self-supervised learning requires extra setup during data generation (Godard et al., 2017; Pinto et al., 2016; Schmidt et al., 2016), unsupervised learning can use existing data without any specific data generation constraints. Unsupervised single image methods such as auto-encoders (Kramer, 1991), colorization (Zhang et al., 2016), GANs (Radford et al., 2015), jigsaw (Noroozi & Favaro, 2016), and NCE (Wu et al., 2018) rely on properties of the images themselves and can be applied to arbitrary datasets. However these datasets cannot represent temporal information, nor can they show novel object views or occlusions.

Video data automatically provides temporal cohesion which can be used as additional supervisory signal to learn these phenomena. There is a long history of using videos for low level (Li et al., 2019; Meister et al., 2018; Srivastava et al., 2015) and high-level tasks (Misra et al., 2016; Wang et al., 2019). One of the most common unsupervised setups is using the present to predict the future. The Natural Language Processing community has embraced language modeling as an unsupervised task which has resulted in numerous breakthroughs (Devlin et al., 2019; Mikolov et al., 2013; Radford et al., 2019). However, similar systems applied to unlabeled videos have not revolutionized computer vision. These representations still underperform supervised methods due to several issues. Primarily, neighboring video frames do not change nearly as much as neighboring words in a sentence, so a network which learns the identity function would perform well at next frame prediction. Additionally, words are reused and can thus be tokenized in an effective way whereas images never repeat, especially between two disparate video sources.

To avoid these issues, many have opted for other methods. Anand et al. (2019) use the NCE loss to discriminate between temporally near frames and temporally far frames of ATARI gameplay but do not compare across games. Han et al. (2019) also use the NCE loss and the CPC technique on a 3D-ResNet to learn spatio-temporal features. Similar to our proposal, Tschannen et al. (2020) and Purushwalkam & Gupta (2020) advocates using videos to learn invariances to deformations, color changes, occlusions, and other difficult scenarios which preserve the semantic relationships of the video subjects but drastically change the appearance. These works provide additional evidence that videos offer a richer visual backing as compared with single image methods.

Aside from the NCE approach, others have proposed alternative video training tasks. Misra et al. (2016) shuffle the frames of a video and train a network to predict whether they are correctly temporally ordered. Wang Wang et al. (2019) and Vondrick et al. (2018) use cycle consistency and color as a form of tracking from one frame onto another. Earlier work from Wang & Gupta (2015) uses hand-crafted features to track patches of a video and learn similarities between the patches. Others have explored learning representations from multi-modal inputs provided by video such as optical flow or audio Korbar et al. (2018); Owens & Efros (2018); Piergiovanni et al. (2020); Zhao et al. (2018). Our approach is inspired by these works but focuses on learning a semantic representation of the entire scene based on a single frame from a video. Many of these prior works require multiple frames for their representation, and thus cannot be used on single-image downstream tasks. However, if a network can consistently represent visually dissimilar images from the same video with similar vectors, then not only has it learned how to recognize what is in each image, but it can also represent what might happen in the past or future of that scene.

## 3 Methods

In order to learn a semantically meaningful representation, we exploit the natural augmentations provided by unlabeled videos. In this section, we first outline the dataset generation process. We then describe the learning algorithm used to train our representation.

### 3.1 Dataset

Using ImageNet as a basis for representation learning has shown remarkable success both with supervised pretraining as well as unsupervised learning. However, even without labels, the images of ImageNet have been hand selected and are unnaturally balanced. To improve learned representations using existing techniques may require significantly larger datasets (He et al., 2020), but obtaining

| Dataset | Number of Images (Train) | Number of Videos (Train) | Number of Categories | Mean Image Size |
|---|---|---|---|---|
| ImageNet 1K (Deng et al., 2009) | 1.3 M | 0 | 1000 | (428, 406) |
| YouTube 8M (Abu-El-Haija et al., 2016) | 0 | 3.7 M | 3862 | - |
| Kinetics 400 (Kay et al., 2017) | 0 | 220 K | 400 | - |
| GOT-10k (Huang et al., 2019) | 1.4 M | 9 K | 563 | (1600, 912) |
| Random Related Video Views (Ours) | 0.96 M | 240 K | - | (467, 280) |

Table 1: **Comparison of various image and video datasets.** While we have neither the most images nor the most videos, we provide good diversity between videos which is crucial for learning a strong, generic image representation.

data with similar properties automatically and at scale is not practical. Instead, we turn to unlabeled videos as a source of additional supervision.

In order to train on a diverse set of realistic video frames, we collect a new dataset which we call Random Related Video Views (R2V2). We use the following fast and automated procedure to generate the images in our dataset.

1. Use YouTube Search to find videos for a set of queries, and download the top K videos licensed under the Creative Commons. In practice we use the ImageNet 1K classes.
2. Filter out videos with static images.
3. Pick a random point in the video and extract $T$ images with a gap of $G$ frames between each image. In practice $T = 4$ and $G = 150$.

Using this procedure, we are able to construct R2V2 in under a day on a single machine. Because our dataset is constructed automatically, we can easily gather more data (more frames per video, more videos overall). In this work we limit the scale to roughly that of comparable datasets (see Table 1). More details on data collection are provided in Appendix 1.

### 3.2 NOISE CONTRASTIVE ESTIMATION (NCE) LEARNING

Given a dataset of diverse video frames, we learn a representation which takes advantage of the structure of the data. We choose the Noise Contrastive Estimation technique (Gutmann & Hyvärinen, 2010) which has been popular in many recent works (Bachman et al., 2019; Chen et al., 2020a; He et al., 2020; Hénaff et al., 2019; Hjelm et al., 2019; Misra & Maaten, 2020; Tian et al., 2019; Wu et al., 2018), augmented with temporal supervision.

The standard NCE implementation (used in (Chen et al., 2020a; Bachman et al., 2019; Hjelm et al., 2019)) uses the following procedure. First, a batch of anchor images $A$ is selected. Second, a batch of positive images $P$ is selected, one for each anchor. Positive matches for one example are reused as negatives for the other samples without the need to recompute the features. The NCE loss for a single batch is shown in Equation 1 where $sim(X, Y)$ is any similarity metric between the two inputs. This similarity is in a learned feature space rather than directly in image space, and this learned feature encoding is what we use on downstream tasks. The NCE Loss produces a strong representational encoding for inputs by maximizing the similarity of the encoding of positive pairs while minimizing similarity of negative ones, encouraging the encoder to be discriminative over many possible matches, but general in that it must work on any inputs. As in other works, we use the dot product of the feature embeddings of the data points (as seen in Equation 2) as the similarity metric due to its computational efficiency (Bachman et al., 2019; Chen et al., 2020a; He et al., 2020; Hénaff et al., 2019; Misra & Maaten, 2020; Wu et al., 2018). The similarity is rescaled by a temperature scalar $\tau$ to create peaked softmax distributions.

$$\mathcal{L}_{\text{NCE}} = -\frac{1}{n} \sum_{i=1}^{n} \log \frac{e^{sim(A_i, P_i)}}{\sum_{j=1}^{n} e^{sim(A_i, P_j)}} \tag{1}$$

$$sim(X, Y) = \tau * \frac{f(X)}{\|f(X)\|} \cdot \frac{g(Y)}{\|g(Y)\|} \tag{2}$$

### 3.3 MULTI-FRAME NCE

All recent works perform some sort of transformation on a single image to create Anchor-Positive pairs for the NCE loss (Bachman et al., 2019; Chen et al., 2020a; He et al., 2020; Hénaff et al., 2019;

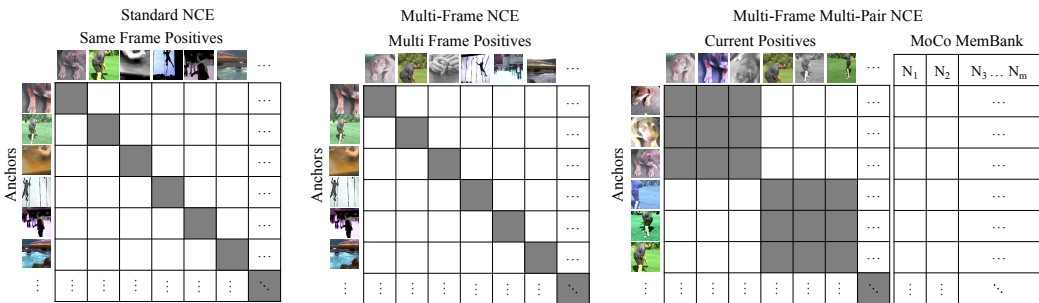

Figure 2: **Left:** Standard NCE using "Same Frame" where all positive pairs come from the same image. **Middle:** NCE using "Multi-Frame" where positive pairs come from the same video. **Right:** Multi-Frame Multi-Pair NCE which uses more than one positive pair per video. The gray boxes indicate the true match pairs. The Memory Bank adds more negatives for each anchor.

Misra & Maaten, 2020; Wu et al., 2018). We refer to this as "Same Frame." We differ from these works by using multiple images from a single video to form our pairs. This allows our network to see truly different views, deformations, similar objects, and larger scene changes. Additionally, this encodes temporal consistency as the semantic contents of a video are unlikely to change suddenly. For example in a video of two cats playing, the camera may focus on one cat, or may even pan to a previously unseen dog, but it is unlikely to pan to a shark. Note that in practice, we select frames with replacement, making our potential pairs a strict superset of those in prior works.

### 3.4 MEMORY BANKS AND MOMENTUM CONTRAST

NCE-based methods benefit greatly from large pools of negatives because this increases the likelihood of finding at least one hard negative for each positive example. In some works (Misra & Maaten, 2020; Wu et al., 2018), negatives are sampled from a large memory bank which was filled with earlier outputs of the network. The NCE loss can be modified to use negatives from prior batches as shown in equation 3 for a memory bank of negatives $N_{1...m}$.

$$\mathcal{L}_{\text{NCE}} = -\frac{1}{n}\sum_{i=1}^{n}\log\frac{e^{sim(A_i,P_i)}}{e^{sim(A_i,P_i)} + \sum_{j=1}^{m}e^{sim(A_i,N_j)}} \tag{3}$$

As the network trains, its output distribution will shift. A potential issue when using a memory bank is the network learns a simple classifier between the current distribution and an old one. Momentum Contrast (MoCo) (He et al., 2020) alleviates this issue by using a quickly updating primary network ($f$ in equation 2) and a slowly updating secondary network ($g$). $f$ is updated based on the NCE loss in equation 3 and $g$ is updated using a momentum rule $g \leftarrow \alpha g + (1-\alpha)f$. The memory bank is filled with previous outputs from the slowly changing network $g$, reducing the likelihood that $f$ will be able to learn a simple recent batch/old batch classifier. For more details, see (He et al., 2020).

### 3.5 MULTI-PAIR NCE

By using a memory bank, we increase the number of negatives without a large computational cost. Yet standard NCE only uses $n$ positive pairs per batch of size $n$. We can further increase the number of positives per batch (while holding batch size constant) by selecting $v$ videos and $k$ samples from each video where $k = \frac{n}{v}$. By computing the pairwise similarity between each pair, we reuse each positive sample $k$ times, resulting in $k^2 v = \frac{n^2}{v}$ positives per batch. Using a simple block-diagonal mask as shown in Figure 2 we can efficiently compute the similarities and NCE loss both between elements of the batch and across a memory bank, achieving a large number of positive comparisons per batch while retaining a large negative size. In practice, we notice no meaningful computational cost to this approach. We refer to this full method using Multi-Pair on video data and the Multi-Frame learning procedure as Video Noise Contrastive Estimation (VINCE). See Appendix 2 for detailed pseudocode.

## 4 EXPERIMENTS

We evaluate our method (**VINCE**) on both single image and temporal tasks by freezing our learned representation and adding a small network (in most cases a single linear layer) for adaptation to new end-tasks. Our learned representation transfers well to a variety of visual tasks, especially tasks which require temporal reasoning. To show this, we compare with multiple strong baselines:

- **MoCo-IN**: Network pretrained on ImageNet (Deng et al., 2009) using the MoCo algorithm.
- **MoCo-R2V2**: Network pretrained on R2V2 using the MoCo algorithm. This uses exactly the same data as VINCE but the Same Frame technique described in Sec. 3.3. We also prevent multiple images from the same video being in the Memory Bank at the same time.
- **Sup-IN**: Network pretrained on fully supervised ImageNet.

To validate the benefits of learning from videos, we additionally compare against an unsupervised image dataset. Since ImageNet itself required time, effort, and money to create, we construct a new static image dataset analogous to our video dataset. Specifically, we search Google Images for the ImageNet synsets and download the top K results for each category. We refer to this as **MoCo-G**). Although other unsupervised video-based methods exist, they have generally underperformed pretrained supervised ImageNet weights or are only applicable to video tasks.

### 4.1 TARGET TASKS

We compare each method on several diverse end-tasks using both ResNet18 (He et al., 2016) and ResNet50 backbones. More training implementation details can be found in Appendix 3. Results for these tasks are shown in Table 2. We train all end-task models using Adam (Kingma & Ba, 2014) and a shared learning rate schedule per task. For each dataset, we use standard data augmentation approaches (crop, flip, color jitter except for tracking). One overall trend to note is the relative gain over MoCo-R2V2. If the single-frame algorithm performed as well as our multi-frame method on temporal tasks, it would indicate minimal temporal understanding. However, the relative gain of VINCE over MoCo-R2V2 on Kinetics (13.85%) and tracking (13.33% and 15.38%) are higher than those on single-frame tasks, showing that our method can incorporate temporal cues.

**ImageNet:** For this task, we use our frozen learned representations, adding a single linear layer after the global average pool. Although none of the methods match the fully supervised performance of ResNet on ImageNet, they do achieve reasonable performance given only single linear layer. It is unsurprising that MoCo pretrained on ImageNet images (MoCo-IN) outperforms our method (0.447 vs 0.400) due to the domain shift between pretrain and end-task. However MoCo pretrained on R2V2 (MoCo-R2V2) suffers nearly a $2\times$ drop in accuracy (8.9%) compared to our method (4.7%), indicating that pretraining on multi-frame matching provides a clear benefit over single frame pairs. Our method, which has never seen an image from ImageNet before, still learns a representation which generalizes well to this new type of data. Even drawing images from a similar class distribution (MoCo-G) does not outperform our method.

**SUN Scenes:** SUN Scenes is a classification dataset in which each image is categorized into one of 397 possible scene types such as airplane cabin, bedroom, and coffee shop. Again we train a single linear layer on top of each pretrained network. This data is quite similar to ImageNet in that each image is well-curated and contains single, unambiguous subjects. As such, the ImageNet fully supervised baseline transfers quite well to SUN Scenes. However VINCE outperforms Sup-IN by a small margin. Again we note a large improvement of VINCE over MoCo-R2V2 (0.495 vs. 0.450). This shows that our method learns to recognize not just the main subject of an image but also the surrounding scene, which requires a richer understanding of the world.

**Kinetics 400:** This dataset consists of 10 second clips from YouTube videos and action labels for each segment. We first download each video and subsample each clip to one frame per second (10 frames per clip). We train a single layer LSTM (Hochreiter & Schmidhuber, 1997) followed by a single linear layer to predict the action category for each segment. Kinetics acts as a crucial test to evaluate whether our model learns temporal cues. VINCE greatly outperform all other methods, whereas traditional baselines such as fine-tuning supervised ImageNet do not adapt well at all. This shows that contrary to popular belief, representations pretrained on ImageNet many not be a good fit for other visual domains, especially on temporal tasks.

| | | Image Classification ImageNet(Deng et al., 2009) | Scene Classification SUN Scenes(Xiao et al., 2010) | Action Recognition Kinetics 400(Kay et al., 2017) | Tracking OTB 2015(Wu et al., 2015) | |
|---|---|---|---|---|---|---|
| | Trained Layer(s) | Linear | Linear | LSTM → Linear | 1x1 Conv | |
| | Metric | Accuracy (Top 1) | Accuracy (Top 1) | Accuracy (Top 1) | Precision | Success |
| **ResNet18** | Sup-IN | **0.696** | 0.491 | 0.207 | 0.557 | 0.396 |
| | MoCo-IN | 0.447 | 0.487 | 0.336 | 0.583 | 0.429 |
| | MoCo-G | 0.393 | 0.444 | 0.313 | 0.551 | 0.413 |
| | MoCo-R2V2 | 0.358 | 0.450 | 0.318 | 0.555 | 0.403 |
| | VINCE (Ours) | 0.400 | **0.495** | **0.362** | **0.629** | **0.465** |
| | Relative Gain over MoCo-R2V2 | 11.91% | 9.93% | 13.85% | 13.33% | 15.38% |
| **ResNet50** | Sup-IN | **0.762** | 0.593 | 0.305 | **0.458** | **0.320** |
| | MoCo-V2-IN (our impl.) | 0.652 | 0.608 | 0.459 | 0.300 | 0.260 |
| | MoCo-R2V2 | 0.536 | 0.581 | 0.456 | 0.386 | 0.299 |
| | VINCE (Ours) | 0.544 | **0.611** | **0.491** | 0.402 | 0.300 |
| | Relative Gain over MoCo-R2V2 | 1.36% | 5.29% | 7.72% | 4.15% | 0.33% |

Table 2: **Comparison of representation performance across a variety of end tasks.** We show improvements over MoCo trained on the same data on all tasks, and outperform MoCo trained on ImageNet as well as supervised pretraining on ImageNet on all tasks but ImageNet itself (and tracking for ResNet50). Each representation uses the same ResNet convolutional backbone, sharing weights across all tasks. Linear (for Kinetics LSTM → Linear) classifiers are the only learned weights for each end task.

**Object Tracking Benchmark (OTB) 2015:** In this dataset, given an initial bounding box around an arbitrary object in the first image, a model must locate the object in the following frames. Difficult tracking datasets like OTB 2015 implicitly require robustness to occlusion and appearance change of the target object, so improved performance would indicate a representation's relative robustness to these challenges. We use the SiamFC (Bertinetto et al., 2016) tracking algorithm on top of our learned representation. SiamFC localizes an object of interest by convolving deep features from the initial bounding box with the current image, similar to template matching (Briechle & Hanebeck, 2001) but in deep feature space. For a more complete explanation, see (Bertinetto et al., 2016). OTB is evaluated using two metrics – precision and success. Precision measures the percentage of frames where the (normalized) center error is less than a certain threshold, averaging over many thresholds. Similarly, success measures the percentage of frames where the Intersection Over Union is more than a range of thresholds.

A representation which works well for SiamFC would have the property that the cross correlation of two images of the same object is high, but the cross correlation of two different objects, or a poorly cropped image of the same object, is low. Pretraining our representations on multiple frames from the same video coincides well with the first objective, however since we use cropped data augmentations, the representations tend to be somewhat invariant to poorly-cropped candidates. Still, the models perform quite well across a variety of difficult tracking instances. Our ResNet18 model transfers significantly better than all other methods indicating a clear benefit to using temporal cues during pretraining. We find, somewhat unexpectedly, that the ResNet18 network fares better than the ResNet50. A likely explanation for this is that the original SiamFC method uses AlexNet (Krizhevsky et al., 2012) with no padding in a fully-convolutional manner. When using ResNet, padding must be applied to keep the outputs the same dimensionality as the inputs. Thus, at training time the network may latch onto zero-padding cues which will not be applicable at test time. This becomes more of an issue the larger the receptive field which is why ResNet50 struggles but ResNet18 is somewhat less affected.

## 4.2 METHOD ABLATION

We validate the effectiveness of Multi-Frame (Sec. 3.3) and Multi-Pair (Sec. 3.5) learning by ablating the number of images from each video used in a batch of comparisons shown in Table 3 (due to computational constraints, we only perform ablations on ResNet18). The first row is equivalent to the procedure done in MoCo (He et al., 2020) i.e. the anchor and positive pairs are two data augmentations of the same image. The second row uses the MoCo procedure as well, however the anchors and positives may be from different images from the same video. The third row uses our Multi-Pair NCE method taking 4 positives and 4 anchors from each video, resulting in 16 positive pairs. A pictorial representation can be seen in Figure 2. Note that when selecting images for row 2 and 3, we use sampling with replacement, making our method a strict super-set of MoCo.

| | Test Task | | | | |
|---|---|---|---|---|---|
| Images Per Video | ImageNet | SUN Scene | Kinetics 400 | OTB 2015 Precision | OTB 2015 Success |
| 1: Same Frame | 0.358 | 0.450 | 0.318 | 0.555 | 0.403 |
| 2: Multi-Frame | 0.381 | 0.478 | **0.361** | 0.622 | **0.464** |
| 8: Multi-Frame Multi-Pair | **0.400** | **0.495** | 0.362 | 0.629 | 0.465 |

Table 3: **Method ablation for VINCE.** We compare using one source image with two augmentations (the standard approach), two different images, or a set of different images. Using Multi-Frame results in a large boost across the board. Multi-Frame Multi-Pair further increases the power of the representation. Note that all methods use the entire dataset, but only Multi-Frame methods use multiple images from a video within one batch.

| | Test Task | | | | |
|---|---|---|---|---|---|
| Pretraining Data | ImageNet | SUN Scene | Kinetics 400 | OTB 2015 Precision | OTB 2015 Success |
| R2V2 IN-Queries | **0.400** | **0.495** | 0.362 | 0.629 | 0.465 |
| R2V2 YT8M URLs | 0.367 | 0.478 | 0.343 | **0.667** | **0.492** |
| Kinetics 400 URLs | 0.368 | **0.494** | **0.390** | 0.612 | 0.456 |

Table 4: **Pretraining data ablation for VINCE.** Each method uses the same training setup but different training data. Since R2V2 uses ImageNet search queries, it outperforms the others on ImageNet. Similarly, pretraining on Kinetics increases performance on the Kinetics task.

We observe across-the-board improvements from both modifications to the MoCo approach. The majority of the improvement comes from using two non-identical frames for matching, but we still gain an additional improvement from using Multi-Pair NCE. Our intuition is that using the Multi-Pair NCE creates gradients that pull each feature towards a global video representation whereas the standard NCE remains more instance-based, only moving a representation in one direction at a time. Thus, we would expect the Multi-Pair NCE features to be more holistically semantic whereas the standard NCE may retain more uniquely identifying features. In fact, we observe a larger performance gap on the more semantic ImageNet and SUN Scene tasks. In contrast, because the Kinetics model uses an LSTM to reason over all input images at once, instance-level features are equally useful as global video features for overall accuracy.

### 4.3 PRETRAINING DATA ABLATION

In Table 4 we explore the effect of different pretraining datasets on end-task performance. For each experiment, we use VINCE but use video data from three different sources: our method of searching ImageNet synset queries, using the URLs from YouTube 8M (Abu-El-Haija et al., 2016), and using the URLs from Kinetics 400 (Kay et al., 2017). Again, we only test ResNet18. Our YouTube8M(YT8M) pretraining data uses the same filtering procedure as R2V2 and contains 5.8 million images from 1.4 million videos. As noted in Table 2 MoCo-IN results, using the same dataset for pretraining as the end-task results in a boost in performance on that specific task but does not indicate that the representation will be better on all tasks. We see this trend is true again when pretraining on Kinetics data. Similarly, since R2V2 uses ImageNet synset for search queries, pretraining on it performs better on ImageNet than the other less-aligned datasets.

In general, this would indicate that given a large enough set of diverse videos, pretraining directly on the unlabeled source data would result in the best performing representation on that data. If this is not possible, then pretraining on a large external source of data may still result in a useful representation. It also indicates that the VINCE method works well on a variety of different pretraining datasets. The increased performance on tracking when using YT8M data could be explained by it simply having access to a larger number of video sources and frames. For generic object tracking, class diversity may be less important than number of samples because the class identity is ignored.

### 5 CONCLUSIONS

In this work we introduced Video Noise Contrastive Estimation, a process for using unlabeled videos to learn an unsupervised image representation. By training on multiple images from the same video, we learn from natural changes such as deformation and viewpoint rather than artificial 2D augmen-

tations. To learn from a large variety of diverse videos, we collect Random Related Video Views in a completely automated fashion. We show across-the-board improvements over the recently proposed MoCo (He et al., 2020) technique on a wide variety of tasks, and we believe Video Noise Contrastive Estimation will extend to other unsupervised methods such as SimCLR (Chen et al., 2020a) and PIRL (Misra & Maaten, 2020) as well as other end-tasks. As representation learning techniques improve, we believe that videos – rather than images – will prove an invaluable resource for pushing the state-of-the-art forward.

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

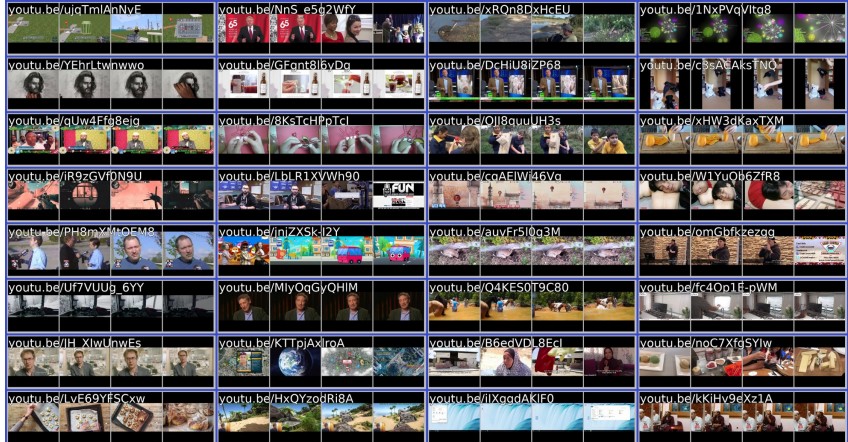

Figure A1: **Sample from Random Related Video Views (train set).**

## APPENDIX 1   DATASET COMPARISON

Random Related Video Views provides nearly 250,000 video links as well as multiple images per video across a broad variety of categories. We show more samples from R2V2 in Figure A1. Videos each have four images 150 frames apart. Each separate video (outlined in blue) lists its corresponding YouTube link. Using ImageNet synsets for search queries provides reasonable visual diversity, but could be substituted with another set of queries. While we acknowledge that using YouTube's Search feature is not truly random, this procedure resulted in significantly more diverse samples than using existing datasets like YouTube8M (Abu-El-Haija et al., 2016) which is heavily unbalanced with unnatural videos.[2] We do no additional data cleaning to ensure that the videos or extracted images actually contain the search term (many do not), nor do we search for "high interest" video segments as in Misra et al. (2016). We also discard the search term itself as a form of supervision. We find that a gap of 5 seconds between each saved image typically results in visually distinct but semantically related images. Too much shorter results in images which are less individually distinct, and too much longer may result in large and unpredictable changes. A sample from each dataset can be seen in the supplemental material.

---

[2]The second most common category in YouTube8M, comprising 540k of the 3.7 million training videos is "Video Game," and the fifth is "Cartoon" with 240k. The category "Minecraft" itself has over 57,000 videos in the dataset whereas the category "Pear" has only 138.

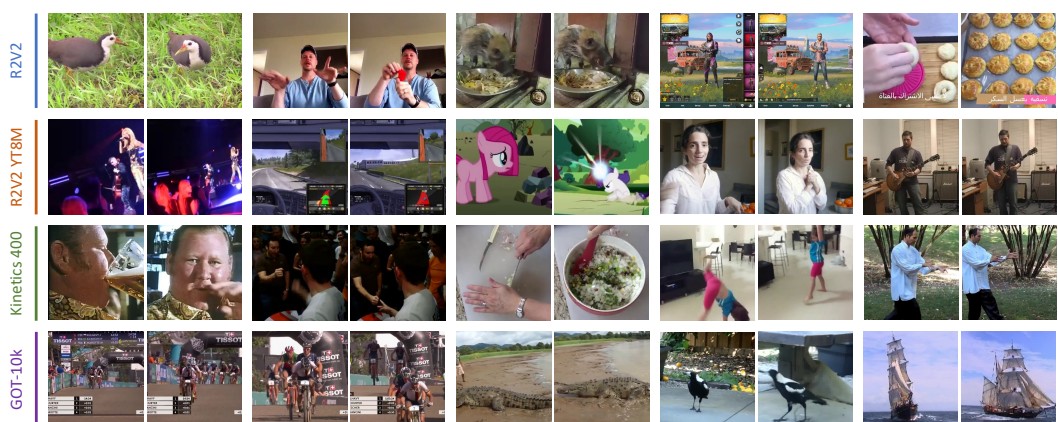

Figure A2: **Random sampling of pairs of images from videos in each dataset.** In GOT-10k, sometimes different video clips are segments from the same original video as seen in the first and second sample. Images are square cropped for visualization purposes only.

We show samples from various existing video datasets in Figure A2. Datasets such as YouTube8M (Abu-El-Haija et al., 2016) and Kinetics400 (Kay et al., 2017). Kinetics contains only videos of humans performing actions. Alternative datasets such as GOT-10k (Huang et al., 2019) provide a comparatively small number of videos but with dense annotations (in GOT-10k's case for object tracking).

## APPENDIX 2    ALGORITHM PSEUDOCODE

The pseudocode below efficiently computes the Multi-Pair NCE for a batch of inputs. Shapes of each input are provided for clarity where:

- $v = $ The number of videos in a batch.
- $k = $ The number of frames per video.
- $n = (v * k) = $ The total number of images in the batch.
- $m = $ The number of elements in the Memory Bank.
- $d = $ The dimensionality of the feature space.

**Algorithm 1** Python-style pseudo code for Multi-Pair NCE.

```
def multi_pair_nce(
        f_output,                                   # [v, k, d] output of f encoder
        g_output,                                   # [v, k, d] output of g encoder
        moco_mem,                                   # [m, d]
        mask,                                       # [n, (n + m)] block diagonal bool
        temperature):                               # [1]

    f_output = f_output.reshape(v * k, d)           # [n, d]
    g_output = g_output.reshape(v * k, d)           # [n, d]
    compare = concatenate((g_output, moco_mem), axis=0)   # [(n + m), d]
    similarities = matmul(f_output, compare.T)      # [n, (n + m)]
    similarities /= temperature
    pos_similarities = similarities[mask]           # [n, k]
    neg_similarities = similarities[!mask]          # [n, (n + m - k)]
    exp_pos_sim = exp(pos_similarities)             # [n, k]
    normalizing_constant = broadcast(               # [n, k]
        reduce_sum(exp(neg_similarities), axis=1),
                    shape(numerator))))
    score = exp_pos_sim / (exp_pos_sim + normalizing_constant)
    loss = -mean(log(score))
    return loss
```

In practice we use the Log-Sum-Exp trick for numerical stability but omit here for clarity. `broadcast` repeats the input until it matches the provided dimensions. `!mask` flips the booleans of each point in the mask.

## APPENDIX 3    VINCE IMPLEMENTATION DETAILS

For training our ResNet18 (He et al., 2016) representations we use the network up to the global average pooling followed by a fully connected layer (512 x 512), Leaky-ReLU and a final embedding layer (512 x 64). The features are L2-normalized and multiplied by $\tau = \frac{1}{0.07}$ as in MoCo (He et al., 2020). We use 8 GPUs, a training batch size of 256 and a MoCo Memory Bank of size 65536 and a $g$-network momentum of 0.999. For multi-GPU training we employ the Shuffle-BN technique shuffling both the anchors and the positives to reduce the correlation between batches filled with multiple images from the same video. We use SGD with a learning rate of 0.03, momentum=0.9, and weight decay=0.0001. All of these hyperparameters are shared with MoCo (He et al., 2020). All methods are trained for approximately 450k iterations. When selecting frames from a video, we pick with replacement. In addition to the natural augmentation, we perform standard data augmentation (color jitter, cropping, flipping) on the inputs. This prevents the network from relying too heavily on shared video statistics like mean frame color. After cropping, all images are resized to $224 \times 224$. It is worth noting that both VINCE and our data (R2V2) can be used with other network architectures or learning algorithms such as AMDIM (Bachman et al., 2019), PIRL (Misra & Maaten, 2020), or SimCLR (Chen et al., 2020a). We choose ResNet18 and MoCo due to implementation simplicity and relatively low computational constraints.

For ResNet50 training, we closely match the hyperparameters in MoCo v2 (Chen et al., 2020b). Specifically, we use the blur augmentation and stronger color augmentation and a cosine annealing

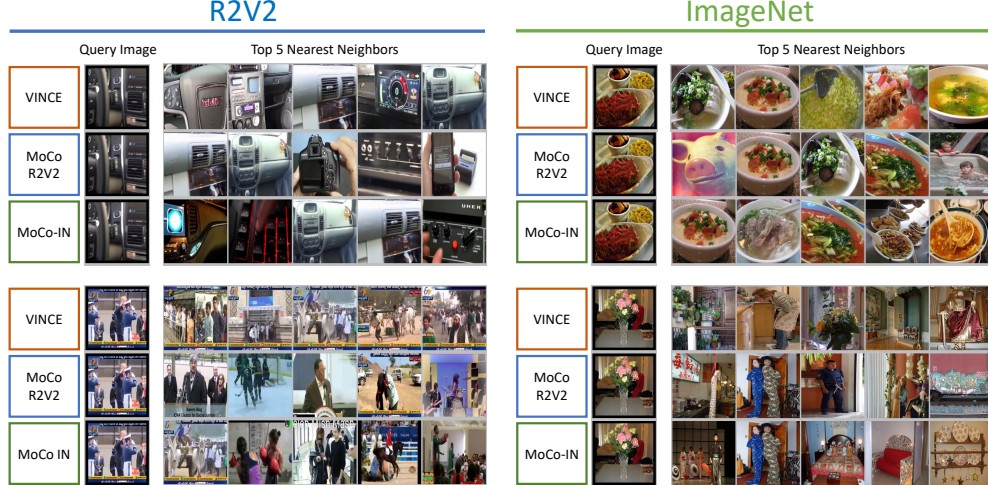

Figure A3: **Nearest neighbor results for a sampling of query images from R2V2 and ImageNet using various models.** VINCE shows a clear understanding of each image and finds highly relevant neighbors.

learning rate for 286,000 iterations (200 epochs of ImageNet or equivalent on R2V2 regardless of the number of unique positives per batch). The batch size we use is 896 with an initial learning rate of 0.105 and an embedding dimensionality of 128. The temperature parameter used was $\tau = \frac{1}{0.2}$. We did not perform any hyperparameter searches for VINCE or R2V2, so these results may be suboptimal.

## APPENDIX 4  QUALITATIVE RESULTS

We additionally provide two qualitative analyses to better understand the success and failure cases of VINCE: Nearest Neighbors, and t-SNE.

### APPENDIX 4.1  NEAREST NEIGHBORS:

We query ImageNet Val and a set of test videos for nearest neighbor matches, taking at most one neighbor per video. We visualize the top 5 neighbors for VINCE, MoCo-R2V2, and MoCo-IN in Figure A3. We observe that VINCE seems to understand the semantics of an image more than MoCo-R2V2 and MoCo-IN. For instance, although MoCo-R2V2 and MoCo-IN find other control panels and buttons in query 1, they do not make the scene-level connection to car interiors as well as VINCE does. Query 2 shows an interesting quirk case of our method. Rather than matching the semantics of the image, VINCE relies on the news logo as a differentiating feature due to its discriminative nature. Each image in VINCE's query 2 results is from a separate video, but from the same news source. For the ImageNet queries, despite never seeing ImageNet inputs during pretraining, VINCE is able to find good matches as well as MoCo-IN which was trained using only ImageNet inputs.

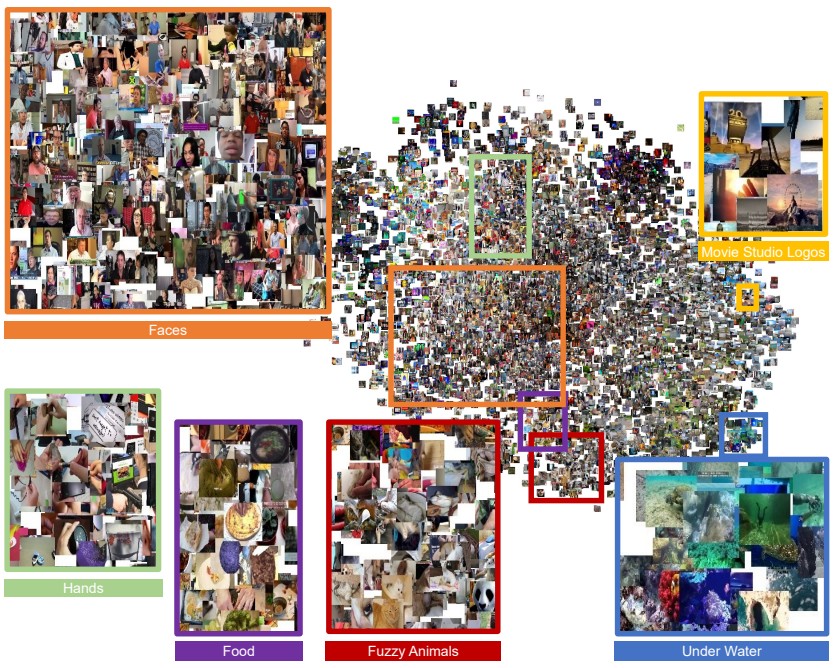

Figure A4: **t-SNE embedding of images from R2V2 test set.**

APPENDIX 4.2    T-SNE:

Using a set of held-out video frames, we project the 64-D embedding space from VINCE to 2D using t-SNE (Maaten & Hinton, 2008) and visualize the formed clusters in Figure A4. Not only does this assist in verifying the quality of the embedding, it also serves as a visual method for evaluating the diversity of the dataset itself. The largest of the clusters seems to be the face cluster. YouTube is full of videos of people looking and talking directly to a camera, and our random subset reflects this pattern. Other interesting, yet unexpected clusters emerge as well such as cats (YouTube loves cats), hands (demo videos), and food (cooking videos).

APPENDIX 5    PRECISION AND SUCCESS PLOTS FOR OTB 2015

We provide full breakdowns of the Precision and Success of each method on OTB 2015 (Wu et al., 2015). The values in the legend correspond to the (mean) area under the curve for each method.

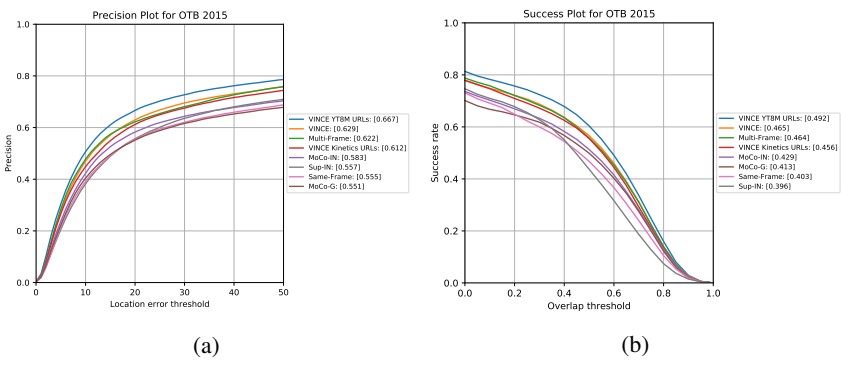

(a)            (b)

Figure A5: **Precision (a) and Success (b) plots for OTB 2015 for various backbones.**

