# OpenReview forum: "Watching the World Go By: Representation Learning from Unlabeled Videos"
_ICLR.cc/2021/Conference — Reject_

### Official Review · AnonReviewer1 · 2020-10-14

**Rating:** 4
**Confidence:** 5

**Review:**

This paper proposes an idea to use NCE for videos where positive/negative training pairs are created by temporally sampling different frames in the video.

The idea is fairly straightforward, there is not anything particularly novel about the approach, other than how the training pairs are created. I think it would have been more interesting and insightful if more ways to generate these pairs were proposed and evaluated. For example, video has many modalities (e.g., RGB, audio, optical flow, etc) in addition to the spatial and temporal dimensions. Exploring ways of creating pairs using multi-modal data would have been more interesting.

Experimentally, there aren't really any comparisons to previous self-supervised learning methods. This is a pretty major weakness as it makes it difficult to understand how well this task is doing. Methods like SimCLR provide >70% accuracy on ImageNet and others do well on video tasks (see missing related works below). Currently, I'm not convinced by the experiments.

The studies comparing different pretraining data and multi-frame vs. single frame are interesting and show the potential of the approach.

There are some missing related works, for example:
- "Cooperative learning of audio and video models from self-supervised synchronization", NeurIPS'18
- "Audio-visual scene analysis with self-supervised multisensory features", ECCV'18
- "Evolving Losses for Unsupervised Video Representation Learning", CVPR'20

These works all provide strong performance on unsupervised video representation learning, yet are not mentioned or compared to.

Overall, I think the proposed idea is not especially novel and the experiments aren't strong enough to show that the simple idea is good. I think there is some potential in the paper, but needs more to be convincing.

---

> ### Author Response · Authors · 2020-11-15
> **Correcting a few misconceptions**
>
> Thank you for your comments on the paper. We appreciate that you think the idea is straightforward. We view this as a strength of our approach (as did SimCLR). We would like to address several of your comments and correct a few misconceptions.
>
> 1. Lack of novelty: True, our paper lacks a large amount of novelty, but we believe the impact shown from our method outperforming other recent popular methods like MoCo V2 gives our paper significant scientific merit.
>
> 2. Multi Modalities and missing citations: You are correct that multi-modal learning has its merits and should be addressed. We will add a short section on multi-modal learning to our related works section. However, multi-modal learning has currently not shown the same performance that simpler single image techniques like MoCo V2 and SimCLR have. Secondly, those specific papers you mentioned only show results on video datasets whereas we show improvements on both single image and video datasets. For more on this, see our response to R3.
>
> 3. Missing Experiments: We chose to compare against two very strong baselines. 1) MoCo V2 which incorporates the MLP suggestion of SimCLR and subsequently outperforms it, and fully supervised pretrained ImageNet weights. Comparing against every new unsupervised method (especially those which have not published pretrained models or code) is a daunting task which we simply did not have the computational resources to do. As noted by the other reviewers, our experiments are actually quite thorough and cover a wide variety of domains in both single image and video tasks.

---

### Official Review · AnonReviewer4 · 2020-10-27
**Learning representations from video sequences, using the Noise Contrastive Estimation proves to yield useful representations**

**Rating:** 8
**Confidence:** 3

**Review:**

The idea of learning representations from video rather than single images is an appealing one with many favorable properties to allow a system to get direct signal on appearance of objects under various natural transformations (occlusion, lighting, etc). Combining instance discrimination ideas of loss based on unlabelled images for which it is known whether they are similar or not, with the idea of curating the images from video is hypothesized to yield learned representations that capture properties enabling improved performance across a variety of single image tasks. The authors create a dataset based on video with positive pairs for noise contrastive estimation, conduct fairly comprehensive experiments and promise to make their newly constructed dataset available. The experiments showcase this type of learned representation outperform alternatives not based on videos on a variety of tasks.

Quality : this seems like a solid paper offering a good intuitive idea with well supported experimental section to show case its relevance.

Clarity : the paper is quite clearly written for the most part. The dataset section 3.1 seems to have an omitted paragraph, please see point 1 below. Section 3.2 is quite lean and does not stand on its own, but relies heavily on previous work omitting much of the essence. I would recommend spending a bit more time on ensuring it is more rigorously written. See for example my comment 2 below.

Originality : the paper is modestly original. It combines two existing ideas - that of using discrimination loss for unsupervised learning of image features, and that of using video based data to allow for rich example of the same data that takes into account real world type transformations. Thus, it is hard to claim more than moderate originality. However,

Significance : the improvements over existing baselines are solid, though I would not categorize them as dramatic. Given the originality is also solid, the overall significance is moderate.

Comments:
1. The dataset generation section is strange, did you omit too much? "We use the following fast and automated procedure to generate the images in our dataset. Using this procedure,..." it almost seems like a few sentences were dropped between the first and second sentences. While the information exists in the appendix, a sentence or two seem to be warranted in the main test.
2. "Gradients flow through the positive pairs" - at this point in the text you have only introduced a loss. The sentence warrants the question of gradients with respect to what? for rigor, and clarity of exposition, this intuition related statement should come after you talk of what is the parameterized aspect of eq (1) wrt which gradients are taken (so after eq 2 is introduced and the idea that the feature representations are captured through learned method, and in particular some reference to the NN you are using. This makes it a bit more complete as a description of the method and presented in a more methodical order.
3. Why would you not remove the option of choosing an anchor and positive as the same image? seems easy to avoid
4. how do you know if the video doesn't contain a shift to a different scene, with different content, thus making the positive actually very different?
5. Since the representation you provide is supposed to learn representations that are somehow more informed about natural transformations (occlusion, lighting) - is there an experiment you can conceive of to test this specific hypothesis? i think it would both be interesting, and also give insight on whether this is indeed what is being learned. This might also make this representation useful for other types of tasks that are not looked at in the paper that I would encourage the authors to explore.

---

> ### Author Response · Authors · 2020-11-14
> **Clarity and clarifications**
>
> Thank you for your thorough and encouraging comments. We appreciate that you have taken into account the significance of our findings in your decision.
>
> 1. Clarity: Frankly, we are embarrassed to see these failures. When worrying about page limits, one often tests removing various sections without losing the core essence of the work. Clearly here we were not careful enough to ensure the proper flow of ideas. We have gone back through the paper thoroughly and made sure all ideas flow naturally and completely. We strive to deliver the ideas of our paper in an intuitive fashion without overwhelming the reader with unnecessary details (retaining completeness through appendices), and we appreciate that for the most part you think we were successful in this.
>
> 2. Removing anchor/positive from the same image pair: This was actually a conscious choice to keep in as it ensures that our method is a strict superset of existing techniques like MoCo. Because we construct our dataset automatically, there may be videos which have poor temporal consistency and are simply too hard to find valuable signal from across different images. This allows us to still use these videos effectively in a single image fashion.
>
> 3. Shift to different scene: We don't, but we have tuned various data collection parameters to attempt to avoid this pitfall. Specifically, the images we choose are separated by 5 seconds each, providing more visual diversity than neighboring frames, but less possibility of total visual independency than 1 minute apart. Furthermore the nature of the potential video pool is somewhat conducive to gathering videos on single subjects. Each video comes from searching a single keyword, and not using videos over 4 minutes in length, so generally they are limited to a single subject matter. We have found this to be true across a small (~1000 videos) but representative set of frames visually inspected by hand. A random sample of frames is available in the appendix.
>
> 4. Direct experiment on natural transformations: This is a great suggestion. We attempted this via the tracking experiment, but could have been more thorough in our exploration of the failure modes of the various models. Robustness to occlusion and appearance change is fundamental to tracking, and as such, many videos in the tracking dataset contain significant examples of these. We will make this point more directly in the paper. The OTB videos can be individually downloaded at http://cvlab.hanyang.ac.kr/tracker_benchmark/datasets.html if you are interested in seeing the actual examples.

---

### Official Review · AnonReviewer3 · 2020-10-28
**Initial review from R3**

**Rating:** 5
**Confidence:** 3

**Review:**

This paper incorporate the popular contrastive with unsupervised learning from video. Specifically, multiple frames from the same video is used as positive pairs and frames from different videos is viewed as negative pair.  The author also proposed a simple and effective ways to collect class-balanced and diverse video frame dataset from Youtube.  The author conducted extensive evaluation experiments on both video recognition and image recognition downstream tasks. Extensive ablation experiments demonstrated the effectiveness  of utilizing multiple frames and the balanced data collection algorithms.

My concerns:
1.  Lack of novelty.  The idea of mapping different frames in one video closer,  predicting other frames in one videos, or maximizing mutual informations of embedding of different frames in one video is widely explored.  And adopting contrastive learning in video unsupervised learning scenario has been done before. (Mentioned in the related work section of your paper too)
2. No comparison against other video based unsupervised learning algorithms.  From my viewpoint, improve over single framed based contrastive learning only proves that your algorithm successfully utilized temporal information encoded in the data and provide limited insights for exploiting more useful information from videos.
3. If you can demonstrate your way of incorporating contrastive learning into video based unsupervised learning offers a non-trivial improvement or have a significant difference with other video based unsupervised learning, the impact of your work will be larger.

Overall, I think this paper is interesting but its contribution is limited.

---

> ### Author Response · Authors · 2020-11-14
> **Novelty and comparison with other video methods**
>
> Thank you for your insightful comments. We appreciate you noting that our experiments were thorough and extensive.
>
> 1. Lack of novelty: Unsupervised learning is a well-explored problem, and the NCE approach has been used successfully in many other works. However prior work with NCE has been limited to single image analysis. We are the first to adapt the NCE approach to video inputs which for the use of single image analysis.
>
> 2. No comparison of other video-based approaches: This is a gap in our experiments, however not as large as you might think. The vast majority of unsupervised video techniques have only been tested on downstream video tasks, and some techniques are not even applicable on single image tasks. Furthermore, with the recent popularity in the NCE approach on single images, a large body of work has come out which essentially states that single images are enough to learn good representations. We refute this claim by showing that even on single image downstream tasks, using videos for pretraining gives a benefit over using single image techniques. This is the primary conceit of our paper, and as such is the primary focus of our experiments. We do appreciate the suggestion to incorporate more existing unsupervised video techniques, and there may indeed be some out there that outperform our method on various tasks, but a limited computational budget makes this unfeasible.

---

### Decision · Program_Chairs · 2021-01-07
**Final Decision**

**Decision:**

Reject

**Comment:**

This paper was a difficult decision. Overall it seems to be a quality paper, well written and with many experiments, in particular evaluating learned representations across various tasks and datasets. The authors were also quite courteous in their replies which is appreciated. I really like the point the paper makes about video as a natural augmentation and I find that novel amid the recent NCE surge, where most papers rely critically on augmentation. R4 was also very positive about the paper overall concept.

In terms of paper weaknesses two of the reviewers voted for rejection because the paper ignores existing work on contrastive learning from videos. The authors rebuttal is that they are the first evaluating on images, not on videos. All reviewers also point out limited technical novelty, which the authors acknowledge. Finally, R1 is not very confident about the experiments.

Overall, and after calibration, the appropriate recommendation seems to be rejection.